# The squeezed dark nuclear spin state in lead halide perovskites

E. Kirstein [1] ✉, D. S. Smirnov [2] ✉, E. A. Zhukov [1], D. R. Yakovlev [1], N. E. Kopteva [1], D. N. Dirin[3], O. Hordiichuk[3,4], M. V. Kovalenko [3,4] & M. Bayer [1]

Coherent many-body states are highly promising for robust quantum information processing. While far-reaching theoretical predictions have been made for various implementations, direct experimental evidence of their appealing properties can be challenging. Here, we demonstrate optical manipulation of the nuclear spin ensemble in the lead halide perovskite semiconductor FAPbBr$_3$ (FA = formamidinium), targeting a long-postulated collective dark state that is insensitive to optical pumping after its build-up. Via optical orientation of localized hole spins we drive the nuclear many-body system into this entangled state, requiring a weak magnetic field of only a few milli-Tesla strength at cryogenic temperatures. During its fast establishment, the nuclear polarization along the optical axis remains small, while the transverse nuclear spin fluctuations are strongly reduced, corresponding to spin squeezing as evidenced by a strong violation of the generalized nuclear squeezing-inequality with $\xi_s < 0.5$. The dark state corresponds to an ~35-body entanglement between the nuclei. Dark nuclear spin states can be exploited to store quantum information benefiting from their long-lived many-body coherence and to perform quantum measurements with a precision beyond the standard limit.

The range of many-body phases in condensed matter is formidably rich: the states underlying the fractional quantum Hall[1,2] and the Wigner crystallization[3–5] effects are just two examples[6]. Quantum coherence is essential for the formation of many of them.

The many-body states of localized spins in solids are arguably one of the best controlled and potentially scalable hardware to that end. For example, an electron spin can create entanglement between multiple photons for one-way quantum computing[7–9]. However, electron and photon spins suffer from fast decoherence or short lifetime, limiting their applicability for quantum information storage and many-body unitary operations. Nuclear spins, on the other hand, are largely isolated from their environment resulting in extended coherence times. Thus, they complement electron spins[10–12], particularly due to possible electron-nuclear spin interfacing[13–17].

Spin-based concepts for solid state quantum computing and quantum metrology rely on many-body entanglement in combination with nuclear spin squeezing. The latter is the ultimate goal of nuclear spin manipulation[18,19]. Almost 20 years ago, a coherent many-body dark nuclear spin state (DNSS) was predicted that can be formed by orienting optically the spins of resident charge carriers interacting with the nuclear spin bath of the host lattice[10,20]. Mathematically, the DNSS is an eigenstate of the raising operator of the total nuclear spin with zero eigenvalue. This is achieved in the experiment by optical pumping: DNSS formation blocks the optical nuclear spin pumping in analogy with optically dark states in ensembles of atoms, which are immune to the light-matter interaction[21]. The DNSS belongs to the class of maximally entangled singlet states, so that it can be exploited not only to suppress the dephasing of localized electron and hole spin

[1]Experimental Physics 2, Department of Physics, TU Dortmund, 44227 Dortmund, Germany. [2]Ioffe Institute, 194021 St. Petersburg, Russia. [3]Laboratory of Inorganic Chemistry, Department of Chemistry and Applied Biosciences, ETH Zürich, 8093 Zürich, Switzerland. [4]Laboratory for Thin Films and Photovoltaics, Department of Advanced Materials and Surfaces, Empa - Swiss Federal Laboratories for Materials Science and Technology, 8600 Dübendorf, Switzerland. ✉e-mail: erik.kirstein@tu-dortmund.de; smirnov@mail.ioffe.ru

qubits[22,23], but also for storage of quantum information[10,11] or for quantum metrology applications[24]. The aim of our study is to experimentally demonstrate the formation of a dark nuclear spin state in solid state for which lead halide perovskite semiconductors are a suitable platform.

The DNSS is characterized by destructive interference of the nuclear spin amplitudes in the collective transverse components, which leads to their strong suppression without large longitudinal nuclear spin polarization [Fig. 1a]. The destructive interference arises from quantum correlations and entanglement between the nuclear spins, somewhat analogous to the spin structure in antiferromagnetic materials. Realization and demonstration of the DNSS have turned out to be challenging due to threats from the dipole-dipole and quadrupole nuclear interactions. In prior works on III-V semiconductor quantum dots[25–27], where the nuclear spin polarization was measured, a limited polarization value was attributed to DNSS formation, but without proof of suppression of the transverse spin fluctuations[20]. Later, detailed investigations of GaAs quantum dots revealed 80% nuclear polarization without any signatures of DNSS formation, i.e., the nuclear spin state in this system could be described by an effective nuclear spin temperature and the absence of correlations[28]. Correlations between nuclear spins were evidenced for non-thermal nuclear spin states[29], narrowed nuclear spin states[30–32], and nuclear frequency focusing[33]. None of these studies evidenced the DNSS. In a recent study[34], the authors reconstructed the nuclear spin statistics and suggested destructive interference between different nuclear spins. However, a measurement of the transverse spin components, mandatory for DNSS demonstration, is missing.

Here we study the intertwined spin dynamics of nuclei and charge carriers in a FAPbBr$_3$ lead halide perovskite crystal, using tailored optical pump-probe schemes. More specifically, we develop a method for assessing the nuclear spin inertia, which allows us to access the range of weak magnetic fields with milliTesla strengths. We find that excitation with circularly polarized light leads to the disappearance of the transverse nuclear spin fluctuations with simultaneous absence of a significant longitudinal nuclear spin polarization. The combination of these two factors directly supports the destructive interference of nuclear spins and thus the DNSS formation. This many-body correlated state builds up fast on short time scales in the ms-range and shows significant spin squeezing evaluated through the Kitagawa-Ueda parameter[35,36] $\xi_s = 0.48$, corresponding to entanglement of about 35 nuclear spins.

## Results

The exciton resonance in the hybrid organic-inorganic lead halide perovskite FAPbBr$_3$ crystal is located at 2.191 eV. We have measured the time-resolved Kerr ellipticity (TRKE) at this energy. We give basic information on the charge carrier spin dynamics and the photoluminescence spectrum in the Supplementary Note 1A. The nuclear spin polarization has been addressed by the nuclear Overhauser field experienced by the resident hole spins, which we observe in TRKE, recorded at the temperature of $T = 1.6$ K in tilted magnetic field, Fig. 1b. The spin dynamics consists of slow and fast oscillating components related to the hole (h) and electron (e) spin precession about the total magnetic field. The corresponding carrier spin g-factors are $g_h = +0.4$ and $g_e = +2.4$ (Supplementary Note 1A).

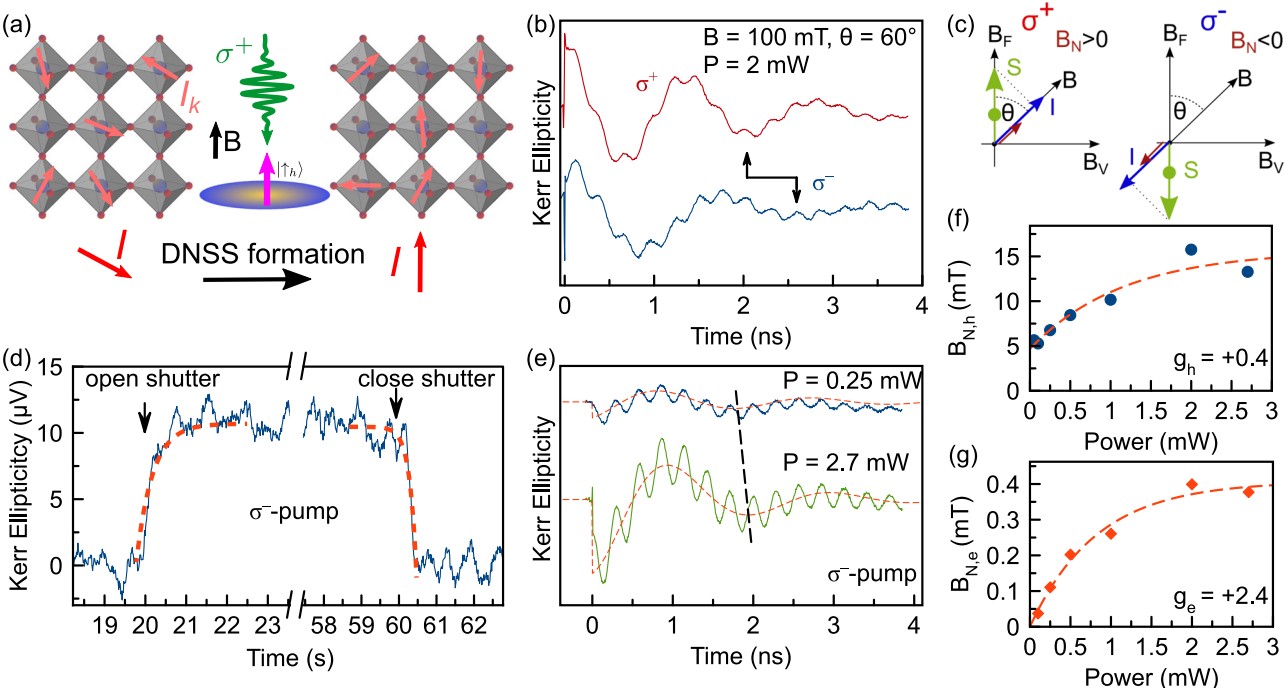

**Fig. 1 | Measurement of the nuclear spin polarization in FAPbBr$_3$. a** Formation of the DNSS: after optical orientation (green light pulse) of a localized hole (magenta arrow), individual nuclear spins $I_k$ (small coral arrows) rotate through the hyperfine interaction, each by the same angle, so that the total nuclear spin polarization $I$ (large red arrow) is directed along the optical axis. **b** Time-resolved Kerr ellipticity dynamics measured for $\sigma^+$ and $\sigma^-$ pump pulse polarization measured at 2.191 eV photon energy, in which the oscillations associated with electron and hole spin precession about the magnetic field are superimposed. Arrows mark the second oscillation minimum in the hole signal after the pump pulse at time zero. **c** Illustration of nuclear spin polarization in tilted magnetic field for $\sigma^+$

(left) and $\sigma^-$ (right) pump polarization. **d** Time-resolved measurement of the Overhauser field rise and decay, showing changes on time scales below 100 ms. Experimental parameters: $T = 1.6$ K, $P = 2$ mW, $B = 0.1$ T, $\theta = 60°$, $\Delta t = 3.6$ ns. **e** Time-resolved Kerr ellipticity dynamics measured at two different pump powers for fixed pump polarization. Dashed line indicates the temporal shift of the hole oscillation for increased pump power. **f, g** Pump power dependence of the Overhauser field for holes and electrons, respectively, extracted from the Kerr ellipticity dynamics, using the hole and electron g-factors of $g_h = +0.4$ and $g_e = +2.4$.

The strong spin-orbit interaction in perovskites provides a selectivity for light absorption of different circular polarizations[37]. As a result, optical spin orientation is feasible for resident electrons and holes, localized at different sites in the sample at low temperatures, which represents a situation similar to that provided by quantum confinement. The angular momentum of a $\sigma^+$ or $\sigma^-$ photon is transferred to the charge carrier spin polarization, and further to the host lattice nuclei through the hyperfine interaction, which leads to build-up of an Overhauser field $\boldsymbol{B}_N$ parallel or antiparallel to the external magnetic field, see Fig. 1c. This field can be directly determined through measurement of the electron and hole Larmor precession frequencies about the total magnetic field, to which the external field and the Overhauser field contribute. With increasing pump power, the Overhauser field rises [Fig. 1e], but saturates at relatively small values of $B_{N,h} = 15$ mT for the holes [Fig. 1f] and $B_{N,e} = 0.4$ mT for the electrons [Fig. 1g]. Similar to other perovskites[38,39], the hyperfine interaction is much stronger for the holes due to the dominant s-type orbital contribution to the Bloch wave functions at the top of the valence band. Thus, we focus on the holes in our study, which dominate the nuclei-related changes of the Kerr ellipticity dynamics. The measured dependence of the Overhauser field on the direction and strength of the external field is presented in the Supplementary Information [Supplementary Fig. 2].

The small measured Overhauser field, as compared to the maximum possible strength of 1.3 T evaluated from the hyperfine coupling constants[40], indicates an unusual nuclear spin statistics which can be highlighted further by measurement of its rise and decay times. Therefore, we first close the pump beam to erase the nuclear spin polarization, and then open the shutter after which we observe almost immediately a nuclear spin polarization, see Fig. 1d. After closing the shutter again, the spin polarization decays on the same time scale as the rise of about 100 ms. Thus, the longitudinal nuclear spin relaxation time $T_{1,N}$ is much shorter than the previously reported $T_{1,N} = 5$ s in $FA_{0.9}Cs_{0.1}PbI_{2.8}Br_{0.2}$[40] and 960 s in $MAPbI_3$[41]. In fact, the variation is limited by the shutter mechanical opening/closing times in

experiment, so that the nuclear spin dynamics occur on times shorter than 100 ms. Moreover, the nuclear spin state is not in quasi-equilibrium, as we show below, so the dynamics can be described by a single longitudinal relaxation time only approximately.

To access this relatively fast nuclear spin dynamics, we developed a technique for measuring the nuclear spin inertia. Similarly to the electron spin inertia technique[42,43], it monitors the Kerr ellipticity signal as function of the longitudinal magnetic field (Faraday geometry) with simultaneous modulation of the pump helicity at different frequencies $f_{mod}$, see Methods. The dependence of the nuclear spin response on $f_{mod}$ reveals the time scale of the nuclear spin dynamics. An example for $f_{mod} = 1$ kHz is shown in Fig. 2a. In zero magnetic field, the hole spin precession in the "frozen" nuclear spin fluctuation leads to the dephasing of 2/3 of the spin polarization on average[44]. Application of a longitudinal magnetic field pushes the hole spin precession frequency towards the $z$-axis, as shown in the inset of Fig. 2a. So the spin dephasing becomes less efficient and the spin polarization increases.

Strikingly, the shape of this polarization recovery curve (PRC) changes with the modulation frequency [Fig. 2b]: at the high frequency of $f_{mod} = 5$ kHz, the PRC resembles a wide dip of Lorentzian shape with the half width at half maximum (HWHM) of 40 mT. With decreasing frequency down to 0.1 kHz, this dip gradually disappears and is replaced by a much narrower dip, having the HWHM of about 1 mT. Generally, a PRC consists of the broad and the narrow component, each of which can be phenomenologically described by a Lorentzian. Their widths depend weakly on $f_{mod}$ [Fig. 2c], but their amplitudes change in opposite ways [Fig. 2d] (Supplementary Note 1E).

The spin relaxation time of a hole is of the order of 1 μs (as measured using the spin inertia method at higher modulation frequencies), so that we attribute the PRC changes with modulation frequency to fast hole spin-induced nuclear spin dynamics occurring on time scales of 1 ms, which is shorter than the typical quantum decoherence time of nuclear spins. Further details will be given in the Theory section, additional pump-power dependence and measurements for $f_{mod} = 0$

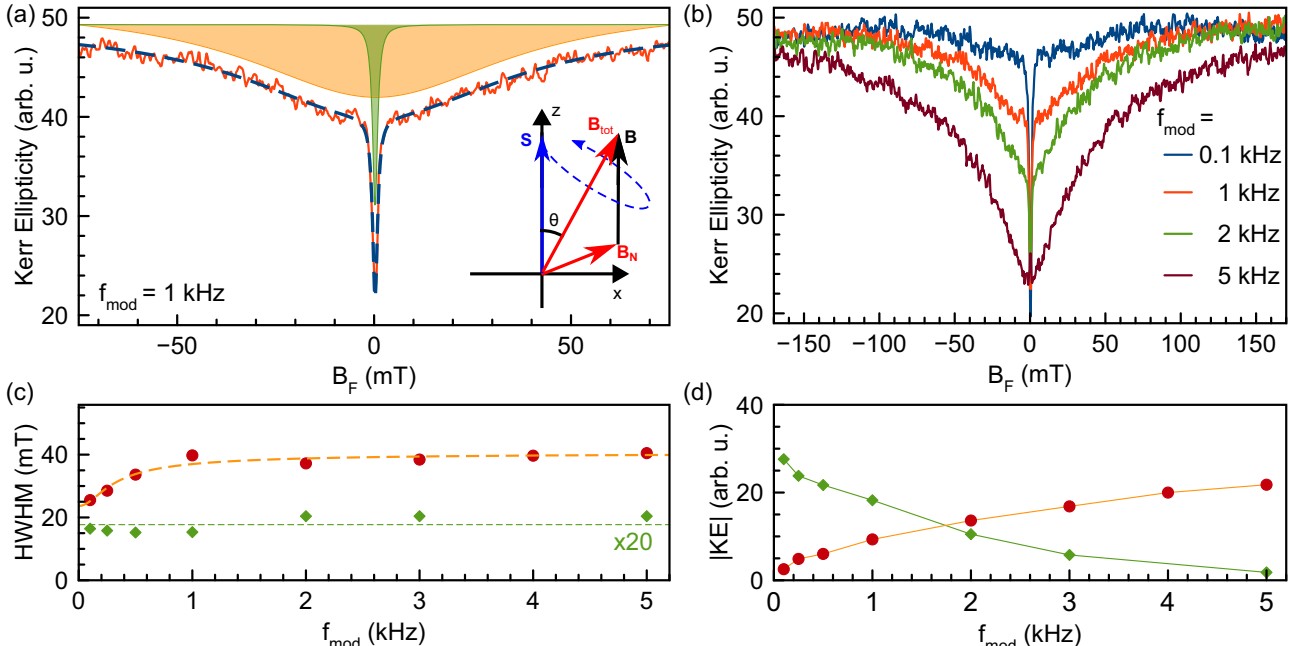

**Fig. 2 | Nuclear spin inertia measurement. a** Representative PRC consisting of a broad (orange shaded) and a narrow (green shaded) component. **b** PRCs at different polarization modulation frequencies. **c, d** Polarization modulation frequency dependence of widths and amplitudes of the broad (red symbols) and narrow (green symbols) PRC components. Lines are guides to the eye. $T = 5$ K. The inset in

(**a**) illustrates the mechanism of polarization recovery: the hole spin (**S**) precesses around the sum (**B**$_{tot}$) of external magnetic field (**B**) and random nuclear field (**B**$_N$), and the smaller the angle $\theta$ between **B**$_{tot}$ and the $z$-axis, the larger the average hole spin polarization. Details on the nuclear spin inertia technique are given in the Methods section.

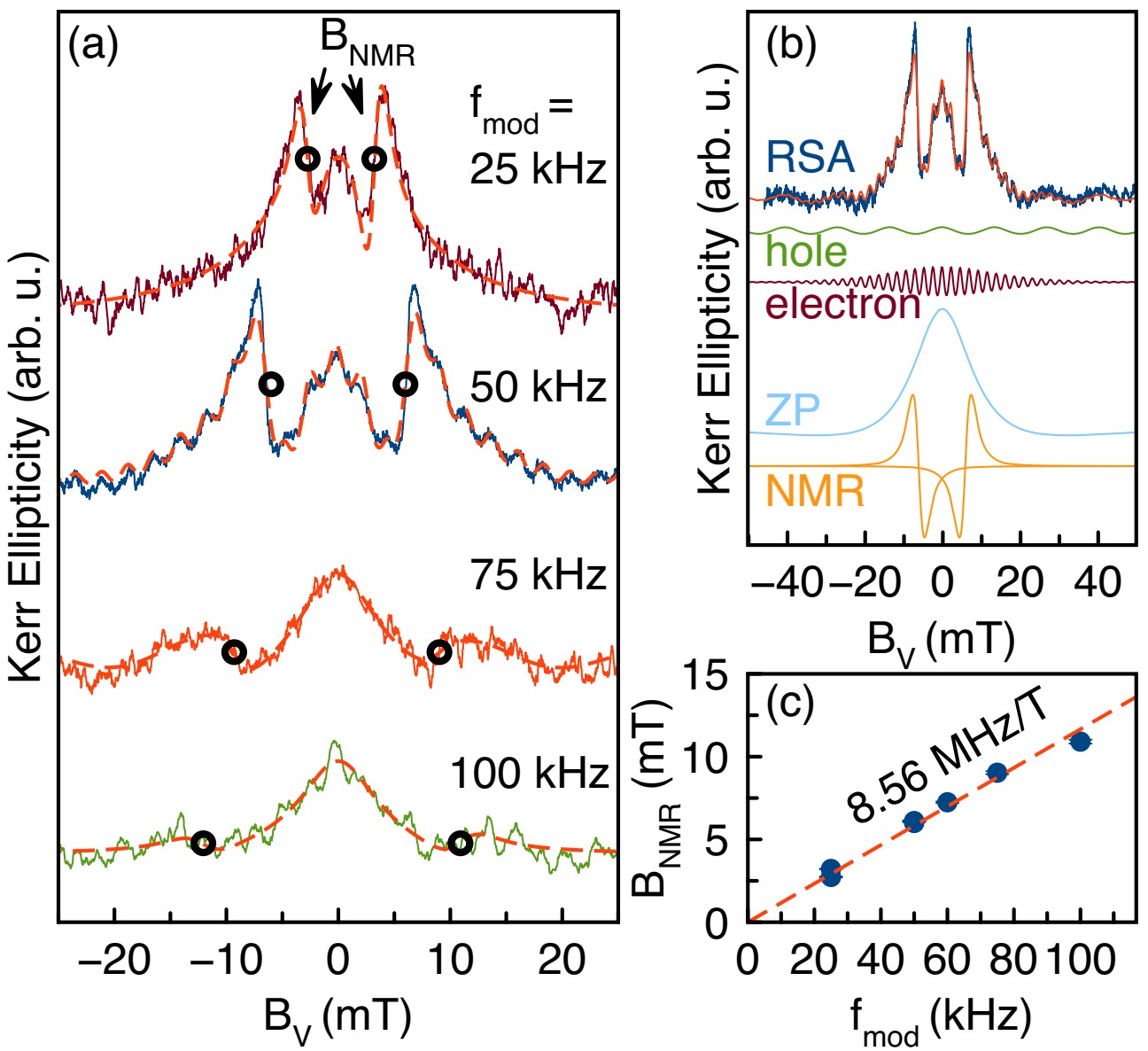

**Fig. 3 | ODNMR measurement. a** Resonant spin amplification signals at different modulation frequencies. **b** Decomposition of the RSA signal at $f_{mod} = 50$ kHz into four components by fitting. From top to bottom: RSA signal (blue) together with the sum of the four components (red), contributions of the hole (green), the electron (dark red), the zero peak [ZP] (light blue), the nuclear magnetic resonances [NMR] (orange). **c** Dependence of the resonance field on the modulation frequency, in agreement with the $^{207}$Pb gyromagnetic ratio. $T = 5$ K.

are presented in the Supplementary Information [Supplementary Figs. 3 and 4].

Further, we identify the specific nuclear isotope dominating the hyperfine interaction. For this, we use the Voigt geometry with the magnetic field perpendicular to the optical axis, where a nuclear magnetic resonance appears if the carrier polarization modulation, equal to the pump modulation $f_{mod}$, matches the nuclear Zeeman splitting[45]. The Kerr ellipticity signal [Fig. 3a] can be separated into an electron-related component and a hole-related component, both representing damped oscillations, in addition a nuclei-related zero field peak and nuclear magnetic resonances (NMR) appear [Fig. 3b]. The latter dominates the signal, their position is given by $B_{NMR} = f_{mod}/\gamma$ with the gyromagnetic ratio $\gamma = 8.56$ MHz/T [Fig. 3a, c]. This value closely corresponds to the gyromagnetic ratio of $^{207}$Pb given by 8.88 MHz/T [see Supplementary Information, Supplementary Table 1]. As in previous measurements of optically detected nuclear magnetic

resonance in perovskites, a small chemical shift may occur due to the presence of spin-polarized holes[40].

When the light polarization modulation frequency is high, the nuclear spin distribution remains in equilibrium. The polarization recovery effect in Fig. 2b at $f_{mod} = 5$ kHz then demonstrates nuclei-dominated hole spin relaxation[46]: In the absence of the external magnetic field, the spin precession in the randomly oriented Overhauser field $\boldsymbol{B}_N$ leads to spin dephasing [Fig. 4a], while application of a longitudinal field $\boldsymbol{B}$ rotates the total magnetic field experienced by the holes, $\boldsymbol{B}_{tot} = \boldsymbol{B}_N + \boldsymbol{B}$, towards the optical axis and suppresses the spin dephasing. Thus, the HWHM of the PRC in this case gives the typical fluctuation of the Overhauser field, $\Delta_B = 40$ mT (Supplementary Note 2A).

The squeezing of the nuclear spin distribution function can be inferred qualitatively from the flattening of the PRC at small modulation frequencies (for $|B| > 1$ mT): Suppression of the hole spin dephasing by the nuclear spin fluctuations unambiguously

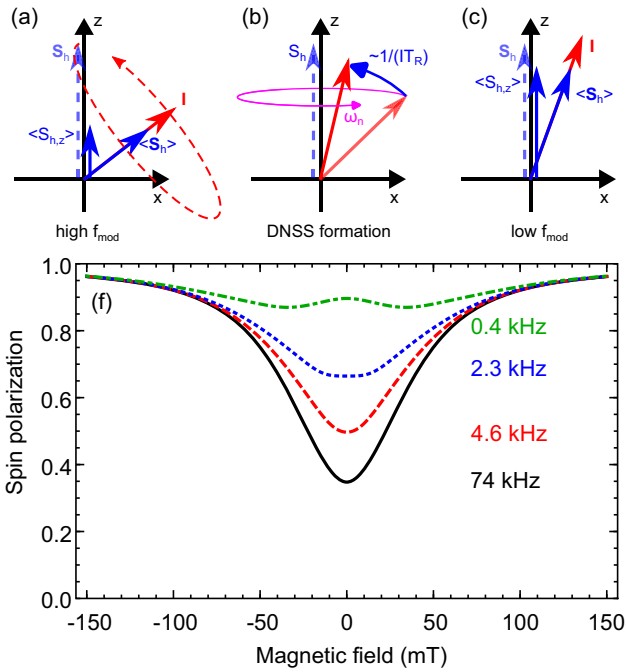

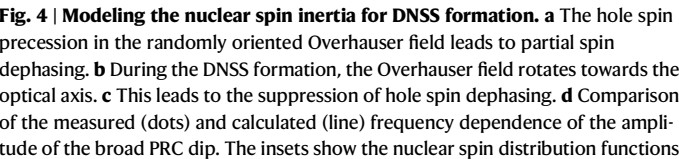

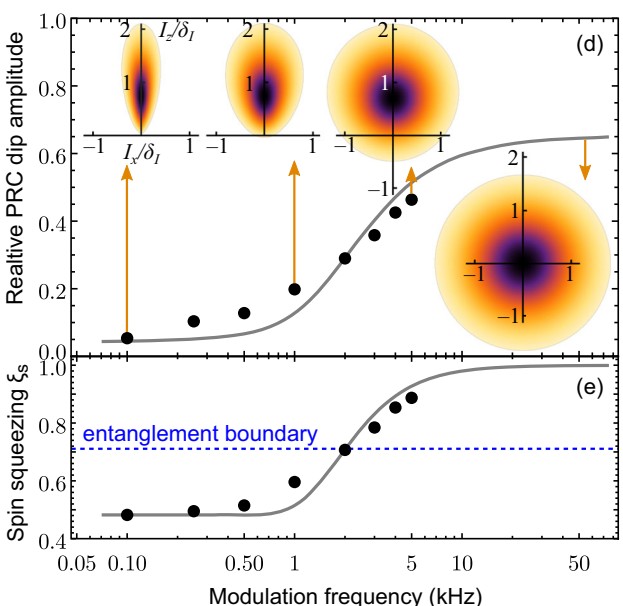

**Fig. 4 | Modeling the nuclear spin inertia for DNSS formation. a** The hole spin precession in the randomly oriented Overhauser field leads to partial spin dephasing. **b** During the DNSS formation, the Overhauser field rotates towards the optical axis. **c** This leads to the suppression of hole spin dephasing. **d** Comparison of the measured (dots) and calculated (line) frequency dependence of the amplitude of the broad PRC dip. The insets show the nuclear spin distribution functions at the corresponding modulation frequencies normalized to their maxima, evidencing the spin squeezing in transverse direction. **e** Nuclear spin squeezing parameter $\xi_s$ calculated from the experimental data (dots) and calculated numerically (line). States below the blue dashed line show entanglement. **f** Calculated frequency dependence of the PRC.

demonstrates alignment of the total nuclear spin fluctuations along the optical $z$-axis, see inset in Fig. 2a. But since the average Overhauser field of 15 mT is considerably smaller than its typical fluctuation $\Delta_B$, this unavoidably leads to the conclusion of reduced transverse fluctuations of the total nuclear spin. The corresponding nuclear spin distribution functions are shown in the insets in Fig. 4d. Below we quantitatively describe the spin squeezing in the "box model" and demonstrate its qualitative validity in Supplementary Note 2E. The spin squeezing arises from the destructive interference between nuclear spins due to their quantum correlations, and is exactly in line with the predictions for the DNSS formation[20].

## Theory

The established concept of dynamic nuclear spin polarization relies on the assumption of an effective nuclear spin temperature and the absence of correlations between the nuclei. In the DNSS, by contrast, quantum correlations and destructive interference in the collective transverse nuclear spin components necessitate a purely quantum mechanical description. Thus, to model the coherent nuclear spin dynamics, we apply the central spin box model described by the following Hamiltonian:

$$\mathcal{H} = A\boldsymbol{I}\boldsymbol{S} + \hbar\Omega_{L,h}S_z. \tag{1}$$

Here $\boldsymbol{I}$ is the total nuclear spin in the hole localization volume, $\boldsymbol{S}$ is the hole spin, $A$ is the hyperfine coupling constant of the holes, $\Omega_L$ is the hole Larmor precession frequency in the external magnetic field applied along the optical $z$-axis, and $\hbar$ is the reduced Planck constant. The spin precession about the nuclear field is included in the renormalization of $\Omega_L$.

The total nuclear spin is formed by $N$ individual $^{207}$Pb 1/2 spins $\boldsymbol{I}_k$: $\boldsymbol{I} = \sum_{k=1}^{N} \boldsymbol{I}_k$. The number of nuclear spins can be estimated from the PRC width using the hyperfine interaction constant $A_0 = AN/\beta = 33\,\mu\text{eV}$[40] with $\beta = 0.22$ being the natural lead spin

abundance: $N = [\beta A_0/(g_h\mu_B\Delta_B)]^2 \approx 60$. Using the lattice constant of $a_0 = 0.6$ nm, this gives the hole localization length $l = a_0(N/\beta)^{1/3} = 4$ nm, which is similar to other perovskites[38,40]. Since $N \gg 1$, Eq. (1) implies hole spin precession between two pump pulses with the constant frequency $A\boldsymbol{I}/\hbar + \Omega_L\boldsymbol{e}_z$ ($\boldsymbol{e}_z$ is the unit vector along the $z$-axis), which leads to hole spin dephasing in weak fields for a randomly oriented total nuclear spin $\boldsymbol{I}$, close to equilibrium.

Continuous hole spin pumping and its dephasing lead to the transfer of angular momentum to the nuclei, as can be seen from the conservation of the total angular momentum $I_z + S_z$ by the Hamiltonian (1). Simultaneously, the conservation of the absolute value of the total nuclear spin $I$ prevents the build-up of a nuclear spin polarization larger than $\sim 1/\sqrt{N}$. Therefore, the total nuclear spin is rotated towards the $z$-axis, while its transverse components decrease, but its absolute value does not increase. The collective nuclear spin dynamics driven by the interaction with a single hole spin produces quantum correlations and leads to entanglement between the nuclei. As a result, the DNSS is formed [Fig. 4b], in agreement with the original theoretical predictions[10,20].

The quantum mechanical solution of the box model[47–50] allows us to fit the experimentally measured amplitude of the broad PRC dip as function of the polarization modulation frequency, see Fig. 4d. The modification of the nuclear spin distribution with decreasing modulation frequency is shown in the corresponding insets. The alignment of all nuclear spin fluctuations along the optical axis at low modulation frequencies cancels the hole spin precession [Fig. 4c] and restores the hole spin polarization to the same value as in large magnetic fields, where the hole and nuclear spins become in effect decoupled (the hole spin dephasing by the nuclei is suppressed). Thereby also the whole PRC amplitude rises, as shown in Fig. 4f. The PRC shape here does not exactly coincide with Fig. 2b due to the spread of the parameters of the hole localization centers. From fitting the nuclear spin inertia at zero magnetic field we find the DNSS formation rate (typical frequency of nuclear spin rotation) of $\nu_0 = 3.2$ ms$^{-1}$ (Supplementary Note 2C).

## Discussion

The ratio of the hole spin polarization at a given magnetic field and in saturation (at $B \gtrsim 100$ mT) represents a collective measurement of the total nuclear spin components $\langle(I_x^2 + I_y^2)/I^2\rangle$[46]. This allows us to quantify the suppression of the transverse nuclear spin fluctuations by the Kitagawa and Ueda spin squeezing parameter $\xi_s^2 = 4\langle I_x^2\rangle/N$[19,35] ($\langle I_x^2\rangle = \langle I_y^2\rangle$), which for uncorrelated spins equals to unity. This parameter extracted from the measured PRC amplitude is plotted in Fig. 4e by the dots, and its simulated frequency dependence with the same DNSS formation rate is shown by the solid line (see also Supplementary Note 2D). For almost complete suppression of the broad PRC dip to values below 0.05 at $f_{mod} = 100$ Hz, the spin squeezing parameter is 0.48, which is limited mainly by quantum fluctuations of the transverse total nuclear spin components and incomplete hole spin polarization.

The quantum correlations between the nuclear spins suggest their entanglement. This can be shown using the generalized spin squeezing inequality[19,51]

$$\langle I_x^2\rangle + \langle I_y^2\rangle + \langle(I_z - \langle I_z\rangle)^2\rangle \geq M/2. \tag{2}$$

Its violation requires $M$-body entanglement between $N$ nuclear spins, meaning that there are at least $M$ spins, which are entangled with the rest of the ensemble[18,52]. To show its violation, we use the upper boundary for the longitudinal nuclear spin fluctuations, $\langle(I_z - \langle I_z\rangle)^2\rangle \leq N/4$, which yields the lower limit of $\xi_s^{(0)} = 0.71$ for the entangled states. States with $\xi_s < \xi_s^{(0)}$ violate Eq. (2) with $M = N$ and are entangled. The maximum achieved DNSS with $\xi_s = 0.48$ is at least $M = 35$-body entangled. However, our theoretical simulations of the nuclear spin fluctuations in the DNSS suggest even deeper entanglement.

There are good reasons for FAPbBr$_3$ perovskites to be the material system for experimental observation of the DNSS: (i) Lead has either nuclear spin 0 ($^{206}$Pb, $^{208}$Pb with abundance of 77.9%) or 1/2 ($^{207}$Pb with abundance of 22.1%), which excludes quadrupole splitting of the nuclear spin levels due to strain or electric field gradients. (ii) The abundance of nonzero lead spins is relatively low and the elementary cell is relatively large. This suppresses the nuclear dipole-dipole interactions, which threaten the DNSS. (iii) The number of nuclear spins in the hole localization volume $N = 60$ is smaller than in typical quantum dots, so their response is faster and the role of the nuclear spin lattice relaxation is weaker. (iv) The magnetic fields that have to be applied are relatively weak of the order of the fluctuations of the Overhauser field, which accelerates the DNSS formation $\propto 1/B^2$.

These factors result in a peculiar regime of the nuclear spin dynamics, which is very different from the usual dynamic nuclear spin polarization. Namely, the longitudinal and transverse nuclear spin dynamics take place at the same time scale $\nu_0^{-1} \sim 1$ ms. During it, the appearance of the longitudinal nuclear spin polarization and suppression of the transverse nuclear spin fluctuations are not mono-exponential so they cannot be described simply by the times $T_1$ and $T_2$. This nuclear spin dynamics is driven mainly by the hyperfine interaction, while the dipole-dipole interactions and spin-lattice relaxation play a minor role.

Our findings establish lead halide perovskites as a promising platform for exploration and exploitation of intertwined hole and nuclear spin dynamics to excite non-classical collective spin states with quantum correlations. The nuclear spin inertia method is powerful for clearly demonstrating DNSS formation, especially for application to other systems with 1/2 nuclear spins including other perovskites or to unstrained quantum dots. In combination with existing demonstrations of nuclear spin based quantum registers[17,53] and collective spin measurements[34,54], the DNSS may be the most reliable platform for quantum metrology, quantum information storage and processing with solid state spins beyond the standard quantum limit[55], due to the low decoherence of the DNSS, caused by destructive interference in the transverse spin fluctuations.

## Methods

### Growth of FAPbBr$_3$ crystals

The FAPbBr$_3$, with FA being formamidinium, perovskite crystal was grown using the inverse temperature crystallization technique (sample code: OH0071a). In essence, the reactant salts (FABr and PbBr2) were dissolved in a mixture of DMF:GBL (1:1 v/v), forming the precursor solution. By rising the temperature of the solution, the sample crystallized due to retrograde solubility of the perovskite crystals in the chosen solvent mixture, see ref. 56. The studied crystal has a reddish color, see the inset in Supplementary Fig. 1a, with a size of $5 \times 5 \times 2$ mm$^3$.

### Magneto-optical measurements

The sample was placed in a cryostat at a temperature variable from 1.6 K up to 300 K. For $T = 1.6$ K the sample was immersed in superfluid helium, while for temperatures in the range from 4.2 K to 300 K the sample was held in cooling helium gas. The cryostat is equipped with a vector magnet composed of three superconducting split coils orthogonal to each other. This allows us to apply magnetic fields up to 3 T in any direction. All magnetic fields, the light vector, and sample surface normal are set to the horizontal plane. Note that the 3D vector magnet allows precise compensation of the residual fields. Magnetic fields parallel to the light wave vector $\mathbf{k}$ are denoted as $\mathbf{B}_F$ (Faraday geometry) and magnetic fields perpendicular to $\mathbf{k}$ as $\mathbf{B}_V$ (Voigt geometry). The angle $\theta$ defines the tilt of the magnetic field from the Faraday geometry.

### Photoluminescence

The photoluminescence (PL) was excited by a continuous-wave laser with the photon energy of 3.06 eV (405 nm). The emitted light was coupled into a 0.5 m monochromator equipped with a Peltier cooled charge coupled device (CCD) via a fiber.

### Time-resolved Kerr ellipticity (TRKE)

The coherent spin dynamics of electrons and holes interacting with the nuclear spins were measured by a degenerate pump-probe setup, where pump and probe have the same photon energy[57]. A titanium-sapphire (Ti:Sa) laser generates 1.5 ps long pulses with a spectral width of about 1 nm (about 1.5 meV) and pulse repetition rate of 76 MHz (repetition period $T_R = 13.2$ ns). The Ti:Sa laser beam was fed into an optical parametric oscillator with an internal frequency doubling and the output photon energy was adjusted to values around the exciton resonance to meet the maximum of the Kerr rotation signal at 2.191 eV (566 nm). The laser output was split into the pump and probe beams. The probe pulses were delayed relative to the pump pulses by a double-pass mechanical delay line with one meter length. The pump and probe beams were modulated using a photoelastic modulator (PEM) for the probe and an electrooptical modulator (EOM) for the pump. The probe beam was always linearly polarized and amplitude modulated at a frequency of 84 kHz. The pump beam was either helicity modulated between the $\sigma^+/\sigma^-$ circular polarizations, or amplitude modulated with fixed helicity, either $\sigma^+$ or $\sigma^-$, in the frequency range from 0 to 5 MHz. In all cases $f_{mod}$ refers to the helicity modulation frequency. Amplitude modulation can be in effect considered as 0 Hz helicity modulation, as the signal is independent from the bare amplitude modulation frequency. In the experiment typically 20 Hz to 100 kHz helicity and amplitude modulation frequencies were used. The polarization of the reflected probe beam was analyzed in respect of the rotation of its elliptical polarization (Kerr ellipticity) with a balanced photodiode, using a lock-in technique. The lock-in band width was, if not otherwise stated, set to 1.3 Hz noise equivalent power.

## Nuclear spin inertia technique

To measure the DNSS formation rate, we tune the modulation frequency of the helicity modulated pump pulses at low frequencies $f_{mod}$ using an EOM. The average pump power is kept constant. The modulation period should exceed the hole spin relaxation time by far but be comparable to the nuclear spin relaxation time, typically ≤5 kHz. We measure the Kerr ellipticity signal, which is proportional to the hole spin polarization, and scan the longitudinal magnetic field (Faraday geometry) in the range ±150 mT, with a step of 240 μT each 170 ms. At high modulation frequencies, the hole spin drives the nuclear spins quickly in the opposite directions, so the nuclear spin bath remains in the equilibrium. For low $f_{mod}$, the hole drives the nuclear spins almost constantly in one direction, which leads to the DNSS formation, whose feedback we expect to see as a flattening of the PRC. The PRC is always measured at a small negative delay of −10 ps. The transition between these two limits allows us to investigate the DNSS formation rate. The probe beam is modulated meanwhile with a PEM, and the signal is demodulated via tandem modulation using two lock-in amplifiers.

## Data availability

The data on which the plots in this paper are based and other findings of this study are available from the corresponding authors upon request.

## Code availability

The code on which the calculations within this paper are based and other findings of this study are available from the corresponding author upon request.

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

## Acknowledgements
We are thankful to M. M. Glazov for fruitful discussions. This work was supported by the Deutsche Forschungsgemeinschaft through the International Collaborative Research Centre TRR 160 (Project A1). The work at ETH Zürich (O.H., D.N.D, and M.V.K.) was financially supported by the Swiss National Science Foundation (grant agreement 186406, funded in conjunction with SPP2196 through DFG-SNSF bilateral program) and by the ETH Zürich through the ETH+ Project SynMatLab. D.S.S. acknowledges the RF President Grant No. MK-5158.2021.1.2, and the Foundation for the Advancement of Theoretical Physics and Mathematics "BASIS". The theoretical modeling of DNSS formation by D.S.S. was supported by the Russian Science Foundation (grant No. 21-72-10035). M.B. acknowledges the support by the Research Center Future Energy Materials and Systems of the Research Alliance Ruhr.

## Author contributions
D.R.Y., E.A.Z., and E.K. conceived the experiment. E.A.Z., E.K., and N.E.K. built the experimental apparatus and performed the measurements. E.K., D.S.S., N.E.K., and D.R.Y. analyzed the data. D.S.S. provided the theoretical description. D.N.D., O.H., and M.V.K. fabricated and characterized the sample. All authors contributed to interpretation of the data. E.K., D.S.S. and D.R.Y. wrote the manuscript in close consultation with M.B. M.B. acknowledges the support by the Research Center Future Energy Materials and Systems of the Research Alliance Ruhr.

## Funding

## Competing interests
The authors declare no competing interests.
