## [Peer Review File · Nature Communications]

REVIEWER COMMENTS

Reviewer #1 (Remarks to the Author):

The authors answered most of the questions, but I still have doubts about the main issue, namely the logical chain that leads from experimental data to conclusions about dark state formation. For example, in the paragraph starting in page 4 and ending in page 5, the authors say that their observation “unavoidably leads to the conclusion of reduced transverse fluctuations of the total nuclear spin”. I don’t think it’s unavoidable. One can easily construct a counter example. Suppose the longitudinal Overhauser field jumps between two equally probable values: +115 mT and -85 mT. The average Overhauser field would then be 15 mT, but the hole spin would each time experience a strong total field, observed as “flattening of the PRC” in an averaged measurement. While this example is exaggerated, it shows that an alternative interpretation of the data is possible. To sum up, I don’t think this work presents a proper (“smoking gun”) proof for dark nuclear state formation. I would say it’s a “maybe” with a lot more work needed for a decisive demonstration. Nevertheless, it’s a good piece of research with new interesting results on a new material system. The paper is concise and is well written. Hence, I don’t object to publication, provided some final improvements are made.

1. To follow from the above, the authors should give more details about the measurements. In particular, the averaging time to measure each point in Fig. 2(b) and the rate at which the static magnetic field is swept in such measurements.

2. Now that the authors provide the numbers for the nuclear spin coherence time, I have some further questions and comments.

2.1 I'm looking at the penultimate paragraph in page 3, where it says "After closing the shutter again, the spin polarization decays on the same time scale of about 100 ms. This nuclear spin response time is much shorter than the previously reported longitudinal nuclear spin relaxation times of $T_{1N} = 5$ s in ... and and 960 s". What is this "response time", is it the nuclear spin longitudinal relaxation time T_{1N} ? If not, what is the T_{1N} in the studied structure? Please avoid ambiguous and misleading terms such as "nuclear spin response time" and quote longitudinal and transverse times explicitly.

2.2 Assuming T_{1N} is 100 ms, and taking the new estimate for the coherence time of $T_{2N} \sim 120$ ms, this would mean a peculiar regime, where nuclear spin longitudinal and transverse relaxation are of the same scale. This would be fine for a liquid, but not for solids, where $T_{1N} \gg T_{2N}$ is the usual expectation. Is there an explanation to this?

2.3. This relation $T_{1N} \sim T_{2N}$ returns me to my original criticism. There appears to be some big effect, other than dark states, which limits the build up of the net nuclear polarization, causes the reduction in T_{1N} , etc. Some solid-state effect, such as charge fluctuations, is far more plausible than the fragile quantum correlations required for a nuclear dark state. I strongly recommend the authors highlight these nuances in the ANALYSIS AND DISCUSSION section. I think a lot more work is needed to make a firm conclusion about dark state formation. Claiming dark state formation with the present set of data and understanding in my view would be misleading to the reader.

3. "we demonstrate coherent optical manipulation of the nuclear spin ensemble" in the abstract. I think it's an overclaim. Coherent manipulation would require demonstration of Rabi rotations, Ramsey fringes or something of that sort. There is nothing like that shown in the present paper.

REVIEWER COMMENTS AND QUESTIONS

(1.) To follow from the above, the authors should give more details about the measurements. In particular, the averaging time to measure each point in Fig. 2(b) and the rate at which the static magnetic field is swept in such measurements.

Authors: We are sorry for these omissions. The sweep time was set to 1.7 mT/s, resulting in a measurement interval of 240 μ T within 170 ms. And the lock-in was set to a band width filter of 1.3 (Noise Equivalent Power) - 1.1 (3dB) Hz corresponding to a time constant of 71 ms. We have added this information to the methods section.

2. Now that the authors provide the numbers for the nuclear spin coherence time, I have some further questions and comments.

2.1 I'm looking at the penultimate paragraph in page 3, where it says "After closing the shutter again, the spin polarization decays on the same time scale of about 100 ms. This nuclear spin response time is much shorter than the previously reported longitudinal nuclear spin relaxation times of $T_{1N} = 5$ s in ... and and 960 s". What is this "response time", is it the nuclear spin longitudinal relaxation time T_{1N} ? If not, what is the T_{1N} in the studied structure? Please avoid ambiguous and misleading terms such as "nuclear spin response time" and quote longitudinal and transverse times explicitly.

2.2 Assuming T_{1N} is 100 ms, and taking the new estimate for the coherence time of $T_{2N} \sim 120$ ms, this would mean a peculiar regime, where nuclear spin longitudinal and transverse relaxation are of the same scale. This would be fine for a liquid, but not for solids, where $T_{1N} \gg T_{2N}$ is the usual expectation. Is there an explanation to this?

2.3. This relation $T_{1N} \sim T_{2N}$ returns me to my original criticism. There appears to be some big effect, other than dark states, which limits the build up of the net nuclear polarization, causes the reduction in T_{1N} , etc. Some solid-state effect, such as charge fluctuations, is far more plausible than the fragile quantum correlations required for a nuclear dark state. I strongly recommend the authors highlight these nuances in the ANALYSIS AND DISCUSSION section. I think a lot more work is needed to make a firm conclusion about dark state formation. Claiming dark state formation with the present set of data and understanding in my view would be misleading to the reader.

Authors: We agree with these remarks. They strongly relate to each other, so that we find it appropriate to provide a coherent response to the three points.

First of all, we have replaced the term "nuclear spin response time" by "longitudinal nuclear spin relaxation time" T_{1N} . The dynamics shown in Fig. 1(d) are limited by the shutter opening and closing times, so the nuclear spin dynamics develop much faster. From the nuclear spin inertia measurements we determine $T_{1N} \sim 1$ ms.

The nuclear spin dynamics in our system is mainly driven by the hyperfine interaction with localized holes. By contrast, the estimate of T_{2N} in Eq. (S21) describes the nuclear spin relaxation if only the dipole-dipole interaction would act, which is weak, resulting in a T_{2N} of the order of 120 ms. It represents a different mechanism of nuclear spin relaxation, so that this time should not be compared directly to $T_{1N} \sim 1$ ms, but underlines that the dipole-dipole interaction is inefficient and can be neglected for describing the observed dynamics. As a result, the longitudinal and transverse nuclear spin dynamics both are driven by the hyperfine interaction with localized holes.

The nuclear spin state in our system is in general quite complex so that its formation cannot be described by a single exponent. During the formation of the dark nuclear spin state (DNSS), the longitudinal nuclear spin polarization appears on the same time scale during which the transverse nuclear spin fluctuations become suppressed. In this sense, the times T_1 and T_2 are comparable. But the formation (and relaxation) of the DNSS is a complex process which involves the entanglement between multiple nuclear spins, so the usual definitions of T_1 and T_2 do not strictly apply to our case.

This situation is indeed peculiar, so we follow the recommendation of the Referee and comment on it in the Discussion section of the main text.

3. “we demonstrate coherent optical manipulation of the nuclear spin ensemble” in the abstract. I think it’s an overclaim. Coherent manipulation would require demonstration of Rabi rotations, Ramsey fringes or something of that sort. There is nothing like that shown in the present paper.

Authors: We agree, and have removed the word “coherent” to avoid ambiguity. At the same time, the formation of squeezed states necessarily requires preservation of the coherence between nuclear spins. So in fact the nuclear spin manipulation in our system is coherent, but in this specific sense.

To conclude, we believe that the present study provides rather conclusive evidence of the DNSS, but certainly further tests and characterization are required for a comprehensive understanding of the DNSS.